# A double-edged hashtag: Evaluation of #ADHD-related TikTok content and its associations with perceptions of ADHD

Vasileia Karasavva[1]*, Caroline Miller[1], Nicole Groves[2], Andrés Montiel[1], Will Canu[3], Amori Mikami[1]

1 University of British Columbia, Vancouver, British Columbia, Canada, 2 Seattle Children's Hospital, Seattle, Washington, United States of America, 3 Appalachian State University, Boone, North Carolina, United States of America

* vkarasavva@psych.ubc.ca

## Abstract

We aimed to assess the psychoeducational quality of TikTok content about attention-deficit/hyperactivity disorder (ADHD) from the perspective of both mental health professionals and young adults across two pre-registered studies. In Study 1, two clinical psychologists with expertise in ADHD evaluated the claims (accuracy, nuance, overall quality as psychoeducation material) made in the top 100 #ADHD TikTok videos. Despite the videos' immense popularity (collectively amassing nearly half a billion views), fewer than 50% of the claims about ADHD symptoms were judged to align with the Diagnostic and Statistical Manual of Mental Disorders. In Study 2, 843 undergraduate students (no ADHD = 224, ADHD self-diagnosis = 421, ADHD formal diagnosis = 198) were asked about their typical frequency of viewing #ADHD content on TikTok and their perceptions of ADHD and were shown the top 5 and bottom 5 psychologist-rated videos from Study 1. A greater typical frequency of watching ADHD-related TikToks was linked to a greater willingness to recommend both the top and bottom-rated videos from Study 1, after controlling for demographics and ADHD diagnostic status. It was also linked to estimating a higher prevalence of ADHD in the general population and greater challenges faced by those with ADHD. Our findings highlight a discrepancy between mental health professionals and young adults regarding the psychoeducational value of #ADHD content on TikTok. Addressing this is crucial to improving access to treatment and enhancing support for those with ADHD.

## 1. Introduction

Owing to social media, it is easier than ever to access information about mental health concerns like attention-deficit/hyperactivity disorder (ADHD), a condition marked by inattention, hyperactivity, and impulsivity that affects approximately 3 - 7% of adults [1,2]. Unlike more traditional psychoeducational sources, social media centers the perspectives of those with *lived experience,* meaning the first-hand experience of a psychological disorder [3]. Consequently, social media can be a powerful tool for showcasing popular information about mental health, reducing stigma, and helping people connect with a community. Nonetheless,

**Data availability statement:** All data used in this work is freely available on the Open Science Framework: https://osf.io/jzgpt/ The datasets for Study 1 are found here: https://osf.io/nwgqm and https://osf.io/ekfjv The dataset for Study 2 is here: https://osf.io/kz4s8

**Funding:** The author(s) received no specific funding for this work.

**Competing interests:** The authors have declared that no competing interests exist.

the lack of vetting, quality control, and moderation of user-generated mental health content on social media may raise concerns. We examined information about ADHD on the social media platform TikTok. Study 1 noted the characteristics of popular ADHD content on TikTok. Study 2 investigated how young adults with and without self-reported ADHD use TikTok, their evaluation of ADHD-related TikTok content, and how consumption of ADHD-related TikTok content may relate to their perceptions of ADHD.

## 1.1 TikTok as a source of information about mental health concerns

TikTok is a social media platform where users post content in short video format. It has grown rapidly in popularity over the past 5 years, and currently boasts over 50 million active daily users who spend almost an hour on the app every day [4]. Up to two in five Americans prefer TikTok over more traditional search engines, like Google; preference for TikTok is strongest among Gen Z (64%) and Millennials (49%) [5]. Despite TikTok's widespread influence and popularity, it remains the least studied among the major social media platforms [6].

A user-driven platform like TikTok can create an environment where people seeking information about mental health, including ADHD, feel empowered to discuss their experiences, connected, and hopeful [7]. At its best, mental health content on social media from peers with lived experience may combat the scarcity of easily and financially accessible resources from mental health professionals. Some people also turn to digital media such as TikTok because they feel afraid, uncertain, or alone, and have internalized stigma about their symptoms – all of which are barriers to seeking face-to-face support. Finally, traditional biomedical approaches to mental health may alienate some people from historically marginalized communities, who may prefer social media to find content that feels less authoritarian and better tailored to their experiences [8,9].

However, social media platforms, like TikTok, are designed in ways that may not be conducive to effective psychoeducation. Easily digestible, short, and snappy videos created to grab users' attention quickly may make it challenging to prioritize nuance [10,11]. Crucially, the TikTok algorithm, ultimately, aims to extend the time users spend on the platform. To do so, TikTok leverages engagement cues such as viewing time, likes, comments, saves, and shares from previous visits to the platform to ensure the videos served to the user cater to their taste, in a process that can go largely unnoticed by users [12,13]. The human tendency for confirmation bias, by which users preferentially read information that supports their pre-existing beliefs about health issues, while ignoring or harshly evaluating information that contradicts them, may compound this process [14]. Repeated exposure to content that aligns with one's pre-existing beliefs increases the content's perceived credibility and the probability of sharing it, a phenomenon referred to as the echo-chamber effect [15].

Financial incentives exist for content creators who post many videos with high view counts, as this translates to receiving payment from TikTok as well as other monetization opportunities such as merch sales, donations, and sponsorships. Producing research-based psychoeducational material requires time, a resource not always available to content creators incentivized to create multiple videos per week. Access to peer-reviewed research is also difficult and expensive for people outside academia; making videos about anecdotal, first-hand experiences may be more feasible. However, the lack of vetting or fact-checking mechanisms could result in the dissemination of inaccurate or misleading information without accountability. In sum, while social media platforms like TikTok offer excellent accessibility and opportunities for peer support and personal expression, their design may not always facilitate nuanced, research-based content, leaving room for the propagation of misinformation.

## 1.2  Psychoeducation on TikTok

Existing literature underscores the disagreement between health professionals and TikTok content creators about health information on the platform. A recent systematic review concluded that authors of scientific articles find social media misinformation prevalent across many health-related topics, including vaccination, noncommunicable diseases, and eating disorders [16].

Specific to mental health content, others found that #autism and #ADHD fell within TikTok's 10 most-viewed health-related hashtags [6]. Yet, Aragon-Guevara et al. [17] reported that among the most popular TikTok videos providing psychoeducation about autism, 41% were rated by the research team as "inaccurate" (e.g., "You can determine if you are autistic using this simple three-question test") and 32% as "overgeneralized" (e.g., "Autistic adults never want to socialize"). Similarly, Brown and colleagues [18] found that out of 100 videos about autism on TikTok, 41% were classified by psychiatrists and medical trainees as "misleading".

Prior research suggests a similar trend for psychoeducational material about ADHD on TikTok. For example, in recent work, 52% of ADHD-related videos evaluated by a psychiatrist and a psychiatry resident with clinical experience in ADHD were classified as "misleading," and only 21% as "useful" [19]. The authors noted that much of the TikTok content about ADHD highlighted individuals' subjective lived experiences. Although content creators did not necessarily make diagnostic claims, the expression of ADHD symptoms can vary from person to person, and treatments that work for one individual may not apply to everyone with ADHD. Failing to provide this nuance may invertedly contribute to misunderstanding of ADHD and could potentially lead to inappropriate generalizations about the condition.

## 1.3  Downstream implications

Mental health-related social media content could influence users' perceptions of whether they have mental health conditions [20–22]. Self-diagnosis is the process by which individuals, without or in conflict with an assessment from a mental health professional, assign themselves a diagnosis based on information they have encountered. By contrast, a formal diagnosis is when a mental health professional gives a diagnosis, not uncommonly based on the synthesis of several pieces and standardized sources of information from multiple informants and methods. Formal diagnosis often increases access to resources and treatment from mental health professionals; instead, people with a self-diagnosis seem to turn to the large online community for information, support, and comfort [23].

Self-diagnosis may empower users, as it puts the conceptualization of psychological health and problems in the hands of many individuals with lived experience, instead of a few "experts" [7]. However, concerns have been raised that misinformation on social media could lead to self-diagnostic inaccuracies. Some work suggests that, in a variety of mental health conditions, social media posts overgeneralize the symptoms of a disorder or overemphasize personal experience as diagnostic evidence [17,24]. Overgeneralization could lead to viewers thinking that their symptoms are more severe than they are, or that things that are part of normal human experience indicate a disorder. For example, information on TikTok could encourage someone to conflate occasional lapses in concentration, in isolation, as evidence of ADHD [25,26].

Notably, adolescents report higher therapeutic alliance with professionals competent in understanding and discussing social media [27]. Yet misinformation on social media could also undermine trust in mental health professionals or turn people away from research-based treatments [28]. Lorenzo-Luaces et al. [29] studied portrayals of cognitive behavioral therapy

on TikTok, concluding that both negative information/personal experience and direct misinformation are frequently posted by non-professionals and are interpreted by viewers as advice. They speculate that a consequence is a mistrust in empirically supported interventions, which could lead users to try potentially ineffective or even harmful treatments. As another example, a common idea that appears on social media is that early childhood vaccines cause autism, despite this claim being repeatedly refuted by empirical investigations [30]. Many social media posts include language that could be categorized as conspiratorial, or that recommend unsupported or untested vaccine alternatives [31]. Interestingly, people's use of social media for information and support regarding physical health conditions is associated with more switching of doctors ("doctor shopping") and suboptimal patient-provider interactions [32] – although the directionality of the effect is impossible to conclude.

### 1.4  Current study

Previous research highlights potential benefits and downsides of the explosion in mental health information on social media platforms. Across two pre-registered studies, we explored how information about ADHD on TikTok is presented, accessed, and evaluated, and the associations with viewers' perceptions of ADHD. Study 1 examined popular TikTok content about ADHD and characterized the reach of this content (Research Question 1; RQ1), and its alignment with clinical diagnostic criteria and treatment recommendations by mental health professionals (RQ2). We predicted that ADHD-related content on TikTok would be popular but not match with empirically supported diagnostic criteria presented in the *Diagnostic and Statistical Manual of Mental Disorders* [33] or professional treatment recommendations for ADHD (e.g., the American Academy of Pediatrics recommendations) [34].

Study 2 examined how young adults with and without self-reported ADHD typically consume ADHD-related TikTok content (RQ3), how they evaluate this content relative to psychologist raters (RQ4), and the associations between consumption and their perceptions about ADHD (RQ5). We predicted that greater consumption of ADHD-related TikTok content would be more prevalent among people with self-diagnoses of ADHD. We also predicted that consuming more ADHD TikTok content would be related to more positive evaluations of the quality of TikTok as a psychoeducation source for ADHD, to overestimating the prevalence of ADHD in the general population, and to estimating that people with and without ADHD have more severe ADHD symptoms. Finally, we explored whether participants chose to watch a video by a psychologist evaluating ADHD-related TikTok content (RQ6), and if watching TikTok videos and the psychologist video related to changes in their confidence about whether or not they had ADHD (RQ7). Analyses of RQ6 and RQ7 were exploratory.

## 2.  Study 1 method

### 2.1  Video search

We sought out ethical approval for Study 1 from The University of British Columbia. On May 19, 2023, we were informed by The University of British Columbia's BREB that since all videos were publicly available on TikTok no consent was required from the creators or any further ethical approval. This study is pre-registered on the Open Science Framework. We created a new TikTok account for the study and queried the hashtag "#ADHD" using the search bar. We sorted the search results by view count and screen recorded the 100 most viewed videos on a single day: January 10, 2023. Notably, the TikTok algorithm offers users videos based on their past engagement on the app, so searching the #ADHD hashtag might show different results to different users. However, because we focused on the most popular videos, our sample of 100 videos likely represents the typical content users see. We chose the hashtag "#ADHD" instead

of others like "#ADHDTok" or "#livingwithADHD" because it was the most popular hashtag in videos portraying ADHD at the time of data collection. Two videos were excluded because they did not include information about the presentation or treatment of ADHD, despite using the #ADHD hashtag. There were no duplicate or non-English videos in our sample. The sample size of 100 videos was selected based on prior research analyzing ADHD and other health-related videos on TikTok, which found it to be a feasible and representative sample of the videos an average user would watch when searching a key term [19,35,36].

## 2.2  Video metrics

For each video, research assistants recorded the number of views, comments, saves, likes, and shares on the day we screen-recorded it. They further noted each video's length, caption, and hashtags, and any credentials of the content creator that were mentioned in the video (e.g., clinical psychologist, licensed MA-level mental health provider, psychiatrist, other medical doctor, psychology student, life coach, lived experience). Next, research assistants perused the TikTok profile of each video's creator noting their follower count, if they posted other ADHD- or mental health-related videos, any credentials in their bio, and any apparent potential for financial gain (e.g., selling products related to the diagnosis, treatment, or management of ADHD, links to Venmo or Cashapp accounts). This was assessed by noting explicit selling of products related to the diagnosis, treatment, or management of ADHD, or explicit solicitations for money, in the included videos or through links to Venmo or Cashapp accounts on the creators' profiles. Any coding disagreements on these video metrics were resolved by consensus between the first and second authors, two clinical psychology PhD students.

## 2.3  Video assessment

Research assistants noted the number of distinct ADHD-related claims made in each video. The claims were classified as concerning (a) ADHD symptoms (e.g., "*My ADHD makes me do this*"; "*People with ADHD act this way*") or (b) treatment for ADHD (e.g., "*This is an effective treatment for ADHD-related problems*"). Any claims about things other than ADHD were ignored. The second and third authors (a PhD student and a postdoctoral fellow in clinical psychology) verified the number and classification of the claims for each video.

   Because there were many more claims concerning ADHD symptoms relative to treatment (see Results), and because these claims were complex, each symptom claim was scored by the fifth and sixth authors, two licensed clinical psychologists, each of whom has 20 + years of expertise in diagnosing and treating ADHD. The two raters independently assessed if the claim accurately captured a core symptom of adult or adolescent ADHD as characterized in the DSM-5 (Yes/No). If it did, the raters noted which ADHD symptom was depicted, and if the severity and impairment illustrated in the video was a realistic representation of what occurs in ADHD. If the claim did not accurately capture a symptom of ADHD according to the DSM-5, the raters assessed whether the claim described a phenomenon strongly linked to ADHD more so than to other disorders (e.g., working memory deficits). They also scored whether the claim better reflected a different specific disorder (e.g., binge eating disorder), a symptom transdiagnostic across various disorders (e.g., emotional dysregulation), or the normal human experience (i.e., suggesting it could reasonably occur among many without significant psychopathology).

   The two psychologist raters also independently coded whether or not there was nuance, defined as if there was any acknowledgement that the symptom depicted in the video may not apply to everyone with ADHD, or may also apply to someone without ADHD. Finally, the raters gave a global score for each video, assessing whether they would recommend it to others

as an example of psychoeducation to help them understand ADHD (1 = *Would definitely not recommend, worried about harm from misinformation, or there is zero useful information even if nothing is inaccurate*; 2 = *Would not recommend, too many caveats or clarifications would be needed such that they outweigh any benefits of the video, or the video just does not provide enough useful information even if nothing is inaccurate*; 3 = *Would recommend but would need to include significant caveats or clarifications*; 4 = *Would recommend with minor caveats or clarifications*; 5 = *Would definitely recommend, no caveats or clarifications needed*).

Overall, there was 84.8% agreement between the two psychologist raters regarding whether the claim reflected symptoms of ADHD as described in the DSM-5 (Cohen's κ = .686). There was 96.4% agreement among the raters on whether a particular claim reflected a symptom of impulsivity (κ = .687), 94.9% agreement for a symptom of hyperactivity (κ = .739) and 86.7% for a symptom of inattention (κ = .672). The global score (assessing whether the raters would recommend the video as ADHD psychoeducation) showed good inter-rater reliability, ICC = .775, 95% CI:.660 -.851. However, acceptable inter-rater reliability could not be achieved on the items assessing whether the severity of the symptom presented seemed realistic.

The second and third authors, an advanced PhD student in clinical psychology and a postdoctoral fellow who specialize in research and treatment of ADHD, independently coded the claims related to ADHD treatment. They noted the type of treatment and whether they considered it to be empirically supported (Yes/No). Judgments were based on extant literature, particularly randomized controlled trials, meta-analytic evidence, and expert consensus statements/guidelines [37–39]. Finally, the raters coded whether nuance was present, defined as whether the content creator acknowledged that the treatment might not work for everyone with ADHD. The raters had perfect agreement about whether or not the suggested treatment was empirically supported and allowed for nuance. All anonymized data for Study 1 can be found on the Open Science Framework.

## 3. Study 1 results

### 3.1 RQ 1: How popular is ADHD-TikTok content?

A summary of the video metrics is in Table 1. Of the 98 videos meeting inclusion criteria, the view count of eight videos was missing in the screen recordings. In total, the 90 remaining videos amassed 495,729,000 views, demonstrating their popularity. This averaged to 5,470,322 views (*SD* = 6,410,138, *Range*: 30,800 – 33,900,000) per video. Each video received, on average, 984,684 likes, 9,728 comments, 71,302 saves, and 19,911 shares. The 10 most popular hashtags in the video captions were #adhdtiktok, #fyp, #neurodivergent, #adhdawareness, #adhdinwomen, #adhdtok, #adhdcheck, #adhdprobs, #relatable, and #foryoupage. This suggests that creators tag the videos with multiple hashtags related to ADHD, neurodivergence in general, and other more generic hashtags (e.g., #fyp; ForYouPage, TikTok's home page), which may be to maximize their audience. Fig 1 has a word cloud with all hashtags used in video captions.

**Table 1. Metrics for the top 100 videos under #ADHD on TikTok.**

|  | Range | *M(SD)* |
|---|---|---|
| Views | 30,800 – 33,900,000 | 5,470,322 (6,410,138) |
| Likes | 5,452 – 25,000,000 | 984,683 (2,589,666) |
| Comments | 0 – 85,700 | 9,728 (13,925) |
| Saves | 93 – 524,400 | 71,301 (91,590) |
| Shares | 109 – 340,400 | 19,911 (39,166) |

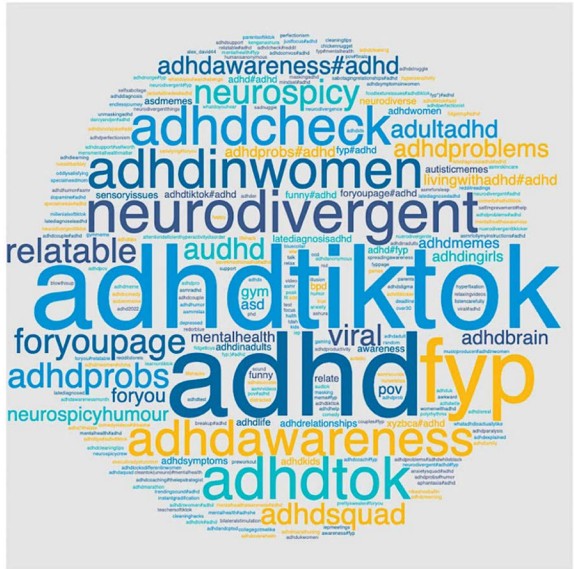

**Fig 1. Hashtag word cloud.**

On average, videos were 38.20 seconds long (*SD* = 36.75, *Range*: 5 – 216) and contained 2.99 claims about ADHD (*SD* = 2.43, *Range* = 1 – 14, *Median* = 2). In 93.9% of videos, the content creator did not refer to any source for their claims. Only 20.4% of creators shared their credentials in the video and 36.7% listed them on their TikTok profile. Of those who did list their credentials, 83.6% cited lived experience, 13.1% reported being life coaches, 1.6% reported being a therapist or counsellor (with no information about license status), and 1.6% reported they were a licenced mental health service provider at the MA level (e.g., licenced clinical social worker, licenced marriage and family therapist). None reported being a licensed mental health service provider at the PhD, PsyD, or MD level (e.g., clinical psychologist, psychiatrist).

Examination of the TikTok profiles of the content creators revealed that 79.2% had posted more than one ADHD-related video, and 44.8% had posted other mental health-related content. Finally, 50% of the content creators promoted products they were selling (e.g., workbooks, fidget spinners, coaching services) or sought financial compensation in another form (e.g., asking users to donate to their Venmo accounts, Amazon Wishlist).

### 3.2 RQ 2: How do psychologist raters evaluate ADHD-TikTok content?

Of the 98 included videos, the vast majority (92 videos; 93.9%) made only claims about ADHD symptoms (e.g., "My ADHD makes me do this"; "People with ADHD act this way"). Four videos (4.1%) made only claims about a treatment for ADHD. Two videos (2.0%) made claims about both ADHD symptoms and treatment.

**3.2.1 ADHD symptom claims.** Using a *Yes* or *No* binary scale, the two psychologist raters scored 48.7% of the claims as accurately reflecting a symptom of adolescent or adult ADHD as characterized by the DSM-5; claims were considered accurate if at least one rater responded '*Yes*'. Of these claims categorized as accurately reflecting an ADHD symptom, 71.3% were judged to portray an inattention symptom, 27.2% a hyperactivity symptom, and 16.2% an impulsivity symptom by at least one of the raters. Because raters often judged a claim to reflect

more than one symptom category (specifically, both hyperactivity and impulsivity), and owing to cases where the two raters disagreed about the type of symptom the claim portrayed, these numbers do not sum to 100%.

Of the claims that were not judged by either rater to depict an ADHD symptom (51.3% of all claims), 5.6% were coded as describing a phenomenon with empirical support for being highly associated with ADHD (and more so than with other disorders, e.g., challenges with executive functioning or working memory), 18.2% as better illustrating a different disorder (e.g., depression, anxiety, eating disorders), 42.0% as a transdiagnostic symptom that could reflect multiple disorders (e.g., emotion dysregulation), and 68.5% as better reflecting normal human experience. The numbers sum to more than 100% owing to disagreement between the two raters.

A video was coded as having nuance if at least one of the two raters endorsed that it was present. However, nuance was rare, with only 4.1% of videos including an acknowledgment that the claims made in the video may not apply to everyone with ADHD, and only 1.4% acknowledging that the symptoms presented may also occur in some people without ADHD. The average global score (assessing whether the raters would recommend the video as ADHD psychoeducation) given by the psychologist raters to the TikTok ADHD videos was 1.78 ($SD$ = 0.82, $Range$ = 1 – 4). Notably, neither rater gave a score of 5 ("*Would definitely recommend, no caveats or clarifications needed*") to any video. See Fig 2.

**3.2.2 ADHD treatment claims.** There were 18 claims about ADHD treatment across the six videos. Of those claims, 44.4% referred to modifying the environment in some way (e.g., a basket for miscellaneous items in each room, white noise, storage next to the exit door). In addition, 44.4% referred to behavioral therapy techniques (e.g., breaking work into smaller tasks, and increasing external rewards). Finally, 11.2% of claims referred to ADHD medication (albeit all did so in a negative manner, implying that it makes one feel bad) and workbooks as a form of ADHD treatment.

The two raters judged 55.6% of the claims to align with an empirically supported ADHD treatment. One claim (5.6%), advertising a workbook, was identified as a product for sale that has little empirical support. Further, 38.9% of the claims were judged as based only on personal experience that does not reflect the empirical literature on the treatment of ADHD. No video acknowledged that the referenced treatment may not work for everyone with ADHD.

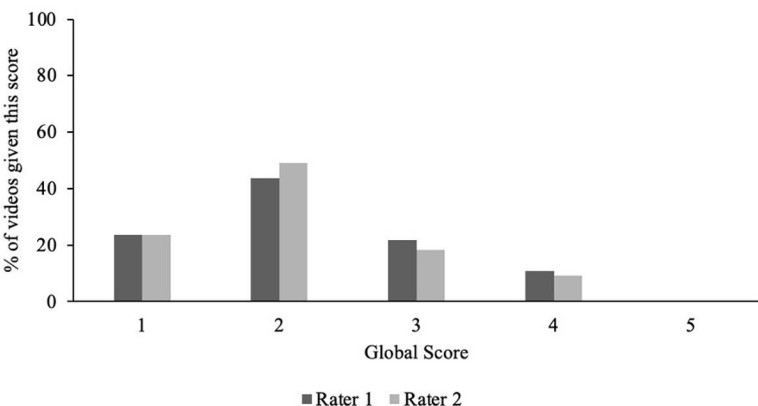

**Fig 2. Distribution of psychologist raters' global scores given to the top 100 #ADHD TikTok videos.**

## 4. Study 2 method

### 4.1 Participants

The sample contained 843 participants (no ADHD = 224, ADHD self-diagnosis = 421, ADHD formal diagnosis = 198) ranging in age between 18-25 years old ($M$ = 20.23, $SD$ = 1.62, $Mdn$ = 22.00), who completed this study in exchange for partial course credit at a public university in Western Canada. Most participants identified as women (79.4%), and the most common responses for racial/ethnic identity were Asian (54.3%) and White/European (35.3%). Please see Table 2 for the demographic breakdown across ADHD diagnostic status groups.

Note. Indigenous = all participants who identified as Aboriginal, Indigenous, Native Canadian, First Nations, Métis, and/or Inuit. Numbers may not sum to 100% of the sample/subsample due to participants being allowed to select multiple options for gender and race/ethnicity and/or due to missing data.

### 4.2 Procedure

This study was approved by The University of British Columbia research ethics board on October 20th, 2023 ([The University of British Columbia BREB number]) and is pre-registered on the Open Science Framework. Data collection took place between January 9th, 2024 and March 25th, 2024. At pre-screening, prospective participants responded "yes" or "no" to two items: "Have you ever been diagnosed with ADHD" and "Do you feel you may have undiagnosed ADHD?". We oversampled for people who answered yes to at least one of these questions. After consenting to the study, participants completed questionnaires about their demographic background and their ADHD diagnostic history and symptoms. Those who

**Table 2. Participant demographics.**

| | Total ($N$ = 843) $N/M$ (SD) | a. ADHD formal diagnosis ($n$ = 198) | b. ADHD self-diagnosis ($n$ = 421) | c. No ADHD ($n$ = 224) | Difference between a and b? | Difference between b and c? | Difference between a and c? |
|---|---|---|---|---|---|---|---|
| Gender | | | | | | | |
| Woman | 669 | 150 | 341 | 178 | $\chi^2$ (4) = **13.02**; ***p* = .01** | $\chi^2$ (4) = 2.66; $p$ = .62 | $\chi^2$ (4) = **15.86**; ***p* = .003** |
| Man | 135 | 27 | 67 | 41 | | | |
| Non-binary | 41 | 21 | 16 | 4 | | | |
| Questioning | 4 | 2 | 1 | 1 | | | |
| Prefer not to disclose | 4 | 1 | 2 | 1 | | | |
| Member of racialized group | | | | | | | |
| Yes | 463 | 102 | 241 | 120 | $\chi^2$ (2) = 1.83; $p$ = .40 | $\chi^2$ (2) = 0.76; $p$ = .69 | $\chi^2$ (2) = 0.31; $p$ = .86 |
| No | 367 | 93 | 174 | 100 | | | |
| Prefer not to disclose | 10 | 2 | 5 | 3 | | | |
| Racial/ethnic background | | | | | | | |
| Indigenous | 20 | 10 | 8 | 2 | $\chi^2$ (7) = **27.22**; ***p* < .001** | $\chi^2$ (7) = 7.13; $p$ = .42 | $\chi^2$ (7) = **27.83**; ***p* < .001** |
| Asian | 458 | 83 | 241 | 134 | | | |
| Black | 18 | 7 | 10 | 1 | | | |
| East Indian | 33 | 5 | 18 | 10 | | | |
| Hispanic/Latinx | 25 | 8 | 11 | 6 | | | |
| Middle Eastern | 39 | 15 | 13 | 11 | | | |
| White | 298 | 94 | 131 | 73 | | | |
| Other | 54 | 15 | 29 | 10 | | | |
| Age | 20.23 (1.62) | 20.51 (1.76) | 20.19 (1.60) | 20.06 (1.52) | ***t*(593) = -2.19, *p* = .03** | $t$(622) = -0.96, $p$ = .34 | ***t*(366.22) = -2.70, *p* = .007** |

answered "no" to both pre-screening questions about ADHD but reported during the study that they had suspected or diagnosed ADHD were removed ($n$ = 33). Therefore, our final "no ADHD" group of $n$ = 224 contains people who answered "no" to both pre-screening questions and confirmed during the study that they did not think they had ADHD. Similarly, people who answered "yes" to one of the pre-screening questions about ADHD but reported during the study that they did not think they had ADHD were removed ($n$ = 211), and not included in the final sample numbers. Among the participants who answered yes to having suspected or diagnosed ADHD in the pre-screening and again in the actual study, we classified them into "formal diagnosis" ($n$ = 198) or "self-diagnosis" ($n$ = 421) groups based on their description of how they received their diagnosis. Those who reported they were assessed by a psychiatrist, family doctor/GP, psychologist, counsellor, or other mental health professional were in the "formal diagnosis" group. Those who reported they were self-diagnosed or that they based their diagnosis on their own research were coded as having a "self-diagnosis".

Participants were informed about the purpose and content of the study and indicated their consent by clicking the "I Agree" button on Qualtrics. Next, the participants who consented completed questionnaires regarding their confidence in their belief that they have or do not have ADHD, their typical engagement with ADHD-related content on TikTok, and other questions about perceptions of ADHD. We then showed participants the top 5 and bottom 5 rated videos from Study 1, based on the global score given to the video by the two psychologist raters. These videos were presented in random order, and participants were not told how they were selected. Participants were asked to evaluate each video after viewing it. After that, they repeated their rating of confidence in their belief that they have or do not have ADHD.

Finally, participants were given the choice to watch a video by one of the psychologist raters from Study 1, explaining why they had scored the 10 videos as either a good or poor representation of ADHD. Participants were informed they would receive full credit even if they did not watch the psychologist video. Those participants who chose to watch the video were asked, after watching it, to again rate their confidence in their belief in whether or not they have ADHD. This concluded the study, and all participants were debriefed and given resources.

Participants were assigned a random ID number, and all anonymized data is available on the Open Science Framework. We did not include the variables on gender identities outside the binary or information about ethnic and racial backgrounds in the shared dataset to reduce the risk of re-identification by linking details such as gender, ethnicity, and age.

### 4.3  Measures

**4.3.1  Demographics and ADHD diagnosis.**  Participants reported their age, gender, and racial/ethnic identity. Participants were also asked how likely it is that they have ADHD on a Likert scale ranging from 1 = *Definitely do not have ADHD* to 5 = *Definitely have ADHD*. Participants who reported any possibility of ADHD were asked how their ADHD diagnosis, if any, was confirmed. Finally, participants were asked to estimate the prevalence of ADHD in the general population, their families, and their friends.

**4.3.2  Confidence in own ADHD.**  Participants rated how confident they are in their ADHD diagnosis (for the formal diagnosis and self-diagnosis groups) or lack thereof (for the no ADHD group) using a single item, on a scale from 1 = *Not at all confident* to 7 = *Very confident*.

**4.3.3  Typical #ADHD TikTok consumption.**  Participants rated the typical frequency that they: (a) view content on TikTok related to ADHD symptoms, (b) view content on TikTok related to ADHD treatment, (c) use the TikTok search bar to look for ADHD-related

content, and (d) organically encounter such content on their ForYou page. Each item was rated on a Likert scale from 1 = *Never* to 5 = *Always*. All items were averaged to create a variable denoting their typical amount of ADHD-related content seen on TikTok with higher scores indicating greater consumption (four items; Cronbach's α =.844).

Next, they rated: (a) how helpful and (b) how accurate the information they typically receive from TikTok about ADHD symptoms is. These questions were repeated regarding the information they receive from TikTok about ADHD treatment. All questions were answered using a Likert scale ranging from 1 = *Not at all* to 5 = *Very*. The four items were averaged to compute a variable assessing participants' perception of existing ADHD-related information on TikTok; greater scores represent more favorable perceptions (four items; Cronbach's α:.877).

Finally, participants reported how much they would recommend the typical ADHD-related TikTok content they see as psychoeducation to help others understand ADHD. Participants made these ratings on the same scale used by the psychologist raters when evaluating the videos in Study 1 (1 = '*Would definitely not recommend, worried about harm from misinformation*' to 5 = '*Would definitely recommend, no caveats or clarifications needed*').

**4.3.4 Adult ADHD self-report scale (ASRS).** The ASRS is based on the World Health Organization Composite International Diagnostic Interview and contains 18 questions assessing the DSM-5 ADHD symptoms as manifested in adults [40]. In our sample, participants were asked to rate the frequency they believed that the average person *with* ADHD (ASRS-with) and the average person *without* ADHD (ASRS-without) struggles with each of the ADHD symptoms on a Likert scale ranging from 1 = *Never* to 5 = *Always*. Both the ASRS-with (18 items; Cronbach's α =.892) and the ASRS-without (18 items; Cronbach's α =.966) showed good internal consistency in our sample.

**4.3.5 Study 1 #ADHD TikTok video evaluations.** After each of the top 5 and bottom 5 psychologist-rated videos from Study 1 that they were shown, participants gave a global score indicating if they would recommend the video as ADHD psychoeducation. Participants responded on the same scale the psychologist raters used in Study 1 to judge the videos (1 = '*Would definitely not recommend, worried about harm from misinformation*' to 5 = '*Would definitely recommend, no caveats or clarifications needed*').

## 5. Study 2 results

### 5.1 Data analysis plan

RQs 3 to 6 were tested using hierarchical multiple regressions. Diagnostic status was dummy coded with the self-diagnosis group as the reference category: (1 = *formal diagnosis*; 0 = *no ADHD or self-diagnosis*) and (1 = *no ADHD*; 0 = *formal diagnosis or self-diagnosis*). In all regressions, demographics (age, gender [0 = *woman*; 1 = *man*]) were entered in the first step, and the two dummy coded variables for ADHD diagnostic status in the second step, to assess their incremental contribution to the criterion variable. Participants' typical amount of ADHD-TikTok content consumption was entered in the third step. The threshold to enter variables in the regression was $p < .005$ and the threshold to remove them was $p \geq .10$ using forward-stepwise regression. Variance inflation factors between all independent variables were less than 10, indicating that the variable intercorrelations were acceptable. Non-binary, two-spirit, and agender people, as well as those questioning their gender identity and those who chose not to disclose their gender ($n = 49$), were not included in the gender analyses. However, these participants were included in all other analyses. A table with their descriptive statistics on all study variables is in Appendix A.RQ7 was tested using repeated measures ANCOVA. All statistical analysis was conducted on SPSS v29.

## 5.2 Descriptive statistics

Table 3 provides descriptive statistics of study variables. All continuous variables in our dataset were normal, as all skewness scores were less than |2| and all kurtosis scores were less than |3|; this was confirmed with visual inspection of QQ plots. Data were visually examined for any outliers using boxplots and histograms and calculating the z-scores for all variables. Cases with a z-score greater than ± 3.29 were considered potential univariate outliers [41]. Next, leverage scores were computed to identify all influential outliers. Because all potential outliers had a leverage score lower than 0.2, they were all retained in the final dataset. The alpha for all analyses was set to .005.

Bivariate correlations between continuous variables are in Table 4. Participants' typical frequency of ADHD-related content consumption on TikTok had significant positive correlations with most other variables of interest, including participants' more favorable perceptions of usual ADHD-TikTok content, evaluations of the top 5 and bottom 5 psychologist-rated TikTok videos shown in the study, and estimates of ADHD prevalence in the population.

## 5.3 RQ 3: How do young adults perceive the ADHD-TikTok content they typically view?

We first examined how ADHD diagnostic status related to participants' typical frequency of consumption of ADHD-related TikTok content, after controlling for participant demographics. Participants with a self-diagnosis of ADHD reported viewing significantly more ADHD-related TikTok content than those with no ADHD ($\beta$ = -0.25, $p$ < .001), and significantly less than those with a formal ADHD diagnosis ($\beta$ = 0.11, $p$ = .002), in their day-to-day life; see Table 5.

**Table 3. Descriptive statistics of study variables.**

|  | Total M(SD) | Skewness | Kurtosis | ADHD formal diagnosis M (SD) | ADHD self-diagnosis M (SD) | No ADHD M (SD) |
|---|---|---|---|---|---|---|
| Confidence in own ADHD |  |  |  |  |  |  |
| T1 | 4.96 (1.40) | -0.44 | -0.20 | 5.99 (1.06) | 4.40 (1.23) | 5.09 (1.43) |
| T2 | 4.89 (1.45) | -0.44 | -0.40 | 5.82 (1.23) | 4.49 (1.28) | 4.80 (1.56) |
| T3 | 5.06 (1.37) | -0.65 | 0.02 | 5.87 (1.08) | 4.62 (1.32) | 5.06 (1.35) |
| ASRS |  |  |  |  |  |  |
| ASRS-with | 3.70 (0.51) | -0.34 | 1.20 | 3.86 (0.48) | 3.74 (0.51) | 3.50 (0.48) |
| ASRS-without | 2.85 (0.81) | 0.66 | -0.41 | 2.82 (0.86) | 2.85 (0.83) | 2.86 (0.71) |
| ADHD-TikTok consumption | 2.39 (0.87) | 0.21 | -0.40 | 2.65 (0.91) | 2.48 (0.83) | 2.00 (0.76) |
| Perception of typical content | 2.38 (0.88) | 0.29 | -0.57 | 2.38 (0.91) | 2.50 (0.86) | 2.18 (0.85) |
| Recommend typical content | 2.53 (0.87) | -0.04 | -0.39 | 2.59 (0.85) | 2.58 (0.89) | 2.39 (0.86) |
| Recommendation top 5 videos |  |  |  |  |  |  |
| Psychologist raters | 3.60 (0.27) | – | – | – | – | – |
| Participants | 2.82 (0.76) | 0.07 | -0.31 | 2.87 (0.78) | 2.88 (0.72) | 2.68 (0.78) |
| Recommendation bottom 5 videos |  |  |  |  |  |  |
| Psychologist raters | 1.10 (0.32) | – | – | – | – | – |
| Participants | 2.32 (0.73) | 0.42 | -0.10 | 2.27 (0.76) | 2.39 (0.71) | 2.23 (0.74) |
| Estimated ADHD prevalence |  |  |  |  |  |  |
| General population | 33.82 (19.50) | 0.62 | -0.40 | 31.25 (19.35) | 37.49 (20.18) | 29.20 (16.89) |
| Family members | 31.56 (25.32) | 0.72 | -0.41 | 45.90 (27.18) | 31.42 (23.83) | 18.13 (17.91) |
| Friends | 36.15 (25.26) | 0.57 | -0.66 | 40.95 (25.80) | 39.23 (25.01) | 25.81 (22.27) |
| Watch psychologist video (% yes) | 51.4% |  |  | 62.4 | 52.3 | 40.0 |

**Table 4. Bivariate correlations between continuous study variables.**

| | 1. | 2. | 3. | 6. | 7. | 8. | 9. | 10. | 11. | 12. |
|---|---|---|---|---|---|---|---|---|---|---|
| 1. ADHD-TikTok consumption | – | | | | | | | | | |
| 2. Perception of typical content | 39*** | – | | | | | | | | |
| 3. Recommend typical content | .27*** | .49*** | – | | | | | | | |
| 4. Recommendation top 5 videos | .08* | .30*** | .38*** | – | | | | | | |
| 5. Recommendation bottom 5 videos | .08* | .24*** | .32*** | .66*** | – | | | | | |
| 6. ASRS (with) | .29*** | .15*** | .12*** | .11** | .05 | – | | | | |
| 7. ASRS (without) | .06 | .01 | -.06 | -.05 | .01 | .20*** | – | | | |
| 8. Estimated ADHD prevalence | .17*** | .14*** | .07 | .06 | .14*** | .08* | -.00 | – | | |
| 9. Confidence in own ADHD (T1) | .06 | .09** | .08* | .07 | .01 | .09** | -.05 | -.08* | – | |
| 10. Confidence in own ADHD (T2) | .12*** | .09** | .13*** | .13*** | .00 | .15*** | -.03 | -.11** | .70*** | – |
| 11. Confidence in own ADHD (T3) | .11* | .12* | .09 | .17** | .07 | .18*** | -.04 | -.082 | .73*** | .88*** |

*Note.* ADHD = Attention Deficit/Hyperactivity Disorder. ASRS = Adult ADHD Self-Report Scale.

$^{*}p < .05$; $^{**}p < .01$; $^{***}p < .001$

**Table 5. Hierarchical multiple regression analyses examining perceptions of ADHD-TikTok content.**

| | TikTok consumption | | | Perception of typical ADHD-TikTok content | | | Recommendation of typical ADHD-TikTok content | | | Recommendation top 5 videos | | | Recommendation bottom 5 videos | | |
|---|---|---|---|---|---|---|---|---|---|---|---|---|---|---|---|
| | β | SE | p | β | SE | p | β | SE | p | β | SE | p | β | SE | p |
| **Step 1** | | | | | | | | | | | | | | | |
| Age | -.02 | .02 | .621 | -.09 | .02 | .006 | -.05 | .02 | .132 | -.06 | .003 | .020 | -.02 | .003 | .386 |
| Gender | -.18 | .07 | <.001 | -.02 | .07 | .528 | -.05 | .07 | .126 | -.04 | .05 | .145 | -.05 | .05 | .084 |
| **Step 2** | | | | | | | | | | | | | | | |
| Formal vs self-dx | .11 | .08 | .002 | -.07 | .08 | .063 | .00 | .08 | .994 | .04 | .06 | .189 | -.08 | .06 | .007 |
| No ADHD vs self-dx | -.25 | .07 | <.001 | .071 | -.07 | .040 | -.01 | .07 | .711 | -.03 | .06 | .253 | -.05 | .05 | .098 |
| **Step 3** | | | | | | | | | | | | | | | |
| TikTok consumption | | | | .36 | .04 | <.001 | .26 | .04 | <.001 | .06 | .03 | .025 | .06 | .03 | .027 |
| $R^2$ | .125 | | | .155 | | | .078 | | | .011 | | | .014 | | |
| F | 27.62 | | <.001 | 28.24 | | <.001 | 12.99 | | <.001 | 3.91 | | .002 | 3.77 | | .002 |

*Note.* Gender is coded as 0 = *woman*; 1 = *man*; formal vs self-dx is the comparison between the ADHD formal diagnosis group with the ADHD self-diagnosis group; no ADHD vs self-dx is the comparison between the no ADHD group with the ADHD self-diagnosis group; TikTok consumption is participants' report of the typical frequency with which they consume ADHD-related TikTok content.

We next examined demographics, ADHD diagnostic status, and typical amount of ADHD-TikTok content consumption as predictors of: (a) participants' perceptions of the ADHD-related TikTok content they typically see, and (b) the likelihood of recommending this TikTok content to others for understanding ADHD. These results are in Table 5. After controlling for demographics, those with a self-diagnosis of ADHD perceived the typical ADHD TikTok content more favorably (as more helpful and more accurate) than those with no ADHD ($\beta$ = -0.07, $p$ = .040), and although not quite meeting the statistical significance criteria, than those with a formal diagnosis of ADHD ($\beta$ = -0.07, $p$ = .063). Importantly, participants who reported a greater typical amount of ADHD-TikTok content consumption also perceived the typical ADHD-TikTok content they viewed as more favorable ($\beta$ = 0.36, $p$ < .001). Only the typical amount of ADHD-TikTok consumption ($\beta$ = 0.26, $p$ < .001) was associated with a greater

likelihood of recommending the typical ADHD-TikTok content they see in the full model after control of covariates.

### 5.4  RQ 4: How do young adults evaluate content compared to psychologist raters?

Table 3 shows participants' evaluations of the top 5 and bottom 5 ADHD-TikTok videos (as judged by our psychologist raters in Study 1), divided by ADHD diagnostic group. On average, participants gave a global score of 2.82 ($SD$ = 0.76) to the top 5 psychologist-rated videos (the psychologists' score was $M$ = 3.60, $SD$ = 0.27), regarding whether they would recommend these videos to others as psychoeducation about ADHD. Participants gave a global score of 2.32 ($SD$ = 0.73) to the bottom 5 videos (the psychologists' score was $M$ = 1.10, $SD$ = 0.32). Thus, participants viewed the top 5 psychologist-rated ADHD-TikTok videos as more worthy of recommendation ($t(824)$ = -23.68, $p$ < .001, *Cohen's d* = 0.61) compared to how participants viewed the bottom 5 psychologist-rated videos. Compared to the psychologist raters, however, participants viewed the top 5 videos *less* favorably ($t(842)$ = 7.71, $p$ < .001, *Cohen's d* = 0.35) and *t*he bottom 5 videos *more* favorably ($t(842)$ = 70.58, $p$ < .001, *Cohen's d* = 3.08).

Using hierarchical regression, we assessed the contribution of demographics, ADHD diagnostic status, and frequency of typical ADHD-TikTok content consumption on the global scores given to the top 5 and bottom 5 psychologist-rated videos (indicating the likelihood that participants would recommend them; Table 5). For the top 5 videos, after adjusting for demographics, there was no significant difference in the global scores given by those with a self-diagnosis versus a formal diagnosis ($\beta$ = 0.04, $p$ =.189) or no diagnosis ($\beta$ = -.03, $p$ =.253). A greater typical frequency of ADHD-TikTok consumption ($\beta$ = 0.06, $p$ =.025) incrementally predicted participants giving higher global scores to the top 5 psychologist-rated videos.

Regarding the bottom 5 psychologist-rated videos in our study, participants with a self-diagnosis gave higher global scores (indicating they were more likely to recommend them) than those with a formal diagnosis ($\beta$ = 0.06, $p$ =.007). No differences emerged between those with a self-diagnosis versus no ADHD ($\beta$ = -0.05, $p$ =.098) in this outcome. Finally, after controlling for the other predictors, those with a greater typical frequency of ADHD-TikTok consumption ($\beta$ = 0.06, $p$ =.027) also gave higher global scores to the bottom 5 psychologist-rated videos.

### 5.5  RQ 5: Does TikTok consumption relate to young adults' perceptions of ADHD?

We used hierarchical regression to examine the contribution of demographics, ADHD diagnostic status, and typical frequency of ADHD-TikTok consumption on the perceived prevalence of ADHD, and on their perceptions of how much the average person with ADHD or without ADHD struggles with the DSM-5 ADHD symptoms. These results are in Table 6.

On average, participants estimated that 33.82% ($SD$ = 19.50%) of the general population, 31.56% ($SD$ = 25.32%) of their family members, and 36.15% ($SD$ = 25.26%) of their friends, could be diagnosed with ADHD (see Table 6). With the understanding that ADHD has a strong heritability index [42] and that individuals with ADHD may be more likely to be friends with others with ADHD [43,44] we focused on the estimated prevalence in the general population. After controlling for demographics, participants with a self-diagnosis gave significantly higher estimates of ADHD prevalence in the general population than did those with no ADHD ($\beta$ = -0.16, $p$ < .001) and those with a formal diagnosis ($\beta$ = -0.14, $p$ < .001). A greater typical frequency of ADHD-TikTok consumption ($\beta$ = 0.13, $p$ < .001) was incrementally

**Table 6. Hierarchical multiple regression analyses examining perceptions of ADHD.**

| | ADHD prevalence in the general population | | | ASRS - with | | | ASRS – without | | | Watch psychologist | | |
|---|---|---|---|---|---|---|---|---|---|---|---|---|
| | β | SE | p | β | SE | p | β | SE | p | OR | SE | p |
| **Step 1** | | | | | | | | | | | | |
| Age | -.03 | .42 | .406 | -.01 | .01 | .687 | -.04 | .02 | .223 | 1.1 | .05 | .050 |
| Gender | -.11 | 1.55 | .001 | -.09 | .04 | .013 | .05 | .07 | .197 | 0.9 | .20 | .717 |
| **Step 2** | | | | | | | | | | | | |
| Formal vs self-dx | -.14 | 1.78 | <.001 | .10 | .05 | .007 | -.04 | .08 | .319 | 1.5 | .18 | .014 |
| No ADHD vs self-dx | -.16 | 1.66 | <.001 | -.16 | .04 | <.001 | .03 | .07 | .505 | 2.5 | .22 | <.001 |
| **Step 3** | | | | | | | | | | | | |
| TikTok consumption | .13 | .84 | <.001 | .22 | .02 | <.001 | .10 | .04 | .009 | 1.07 | .09 | .452 |
| $R^2$ | .074 | | | .133 | | | .013 | | | Nagelkerke $R^2$ = .06 PPV = 59.8% NPV = 57.6% Sens = 59.7% Spec = 57.8% | | |
| F | 12.28 | | <.001 | 23.51 | | <.001 | 1.97 | | .080 | $\chi^2(6) = 33.70$, $p < .001$ | | |

*Note.* Gender is coded as 0 = *woman*; 1 = *man*; formal vs self-dx is the comparison between the ADHD formal diagnosis group with the ADHD self-diagnosis group; no ADHD vs self-dx is the comparison between the no ADHD group with the ADHD self-diagnosis group; TikTok consumption is participants' report of the typical frequency with which they consume ADHD-related TikTok content.

ASRS = Adult ADHD Self-Report Scale.

PPV = Positive Predictive Value; NPV = Negative Predictive Value; Sens = Sensitivity; Spec = Specificity.

All results presented are for the final model. β is the standardized beta coefficient. OR = Odds Ratio.

associated with higher estimates of ADHD prevalence rates in the general population, after controlling for demographics and ADHD diagnostic status.

Regarding expectations about how much the average person with ADHD experiences the DSM-5 ADHD symptoms, participants with a self-diagnosis estimated more symptom severity than did those with no ADHD ($\beta$ = -0.16, $p < .001$), but *less* than those with a formal ADHD diagnosis ($\beta$ = 0.10, $p = .007$). A greater typical frequency of ADHD-TikTok consumption ($\beta$ = 0.22, $p < .001$) was incrementally associated with perceptions that the typical person with ADHD struggles *more* with their symptoms. Only a greater typical frequency of watching ADHD-related content on TikTok was associated with expecting the average person without ADHD to have more severe ADHD symptoms ($\beta$ = 0.10, $p = .009$) after controlling for covariates.

### 5.6 RQ 6: What influences the choice to watch a psychologist evaluate TikTok content?

In total, 51.4% of participants chose to watch the video where one of the psychologist raters from Study 1 explained why they evaluated the ADHD-TikTok videos in the study as either good or poor representations of the disorder. See Table 3. We used a binomial logistic regression to ascertain how demographics, ADHD diagnostic status and frequency of typical ADHD-TikTok consumption relate to the likelihood of opting to watch the psychologist video (yes, no). The Box-Tidwell procedure suggested that all continuous independent variables were linearly related to the logit of each of the dependent variables, and the model was a good fit based on the Hosmer and Lemeshow test of goodness ($p = .581$). Table 6 shows that participants with a self-diagnosis were 1.5 times more likely, and those with a formal diagnosis 2.5 times more likely, to opt-in to the psychologist video than participants with no ADHD. Participants' typical frequency of ADHD-TikTok consumption was not related to the choice to watch the video.

### 5.7 RQ 7: Does watching ADHD-TikTok videos and the psychologist video relate to participants' confidence in their ADHD status?

A one-way repeated measures ANCOVA compared participants' confidence in their ADHD status or lack thereof across three time points: at the beginning of the study (time 1), after participants watched the 10 TikTok videos about ADHD (time 2), and after they watched a video of the psychologist rater evaluating these 10 TikTok videos (time 3). Analyses controlled for the typical frequency of ADHD-TikTok consumption. Recall that the dependent variable assesses participants' confidence that they did have ADHD in the formal diagnosis and self-diagnosis groups and their confidence that they do not have ADHD in the no ADHD group. In this ANCOVA, ADHD diagnostic status was entered as the grouping variable (formal diagnosis, self-diagnosis, no ADHD). Analyses evaluated the main effect of time, the time by ADHD diagnostic status interaction, and the time by frequency of ADHD TikTok consumption interaction.

Mauchly's Test of Sphericity suggested that the assumption of sphericity was violated ($p$ > .05) and the degrees of freedom were adjusted using the Greenhouse-Heisser correction. There was no significant main effect of time, $F(1.69, 605.33) = 0.53$, $p = .588$, partial eta = .001, or a significant interaction between time and frequency of consuming ADHD-TikTok content, $F(1.69, 605.33) = 0.10$, $p = .874$, partial eta > .001. However, there was a significant interaction between time and ADHD diagnostic status, $F(3.38, 605.33) = 4.53$, $p = .003$, partial eta = .025. We followed up this interaction effect with post hoc analysis while applying the least significant difference method to correct for multiple comparisons. The patterns are illustrated in Fig 3. We note that the changes in the means were fairly small and that these analyses only included participants who chose to watch the psychologist video.

Results suggested that participants with a formal diagnosis had significantly higher confidence in their ADHD status than participants with no diagnosis or a self-diagnosis, and their confidence did not appear to change over the study timepoints. By contrast, those in the no

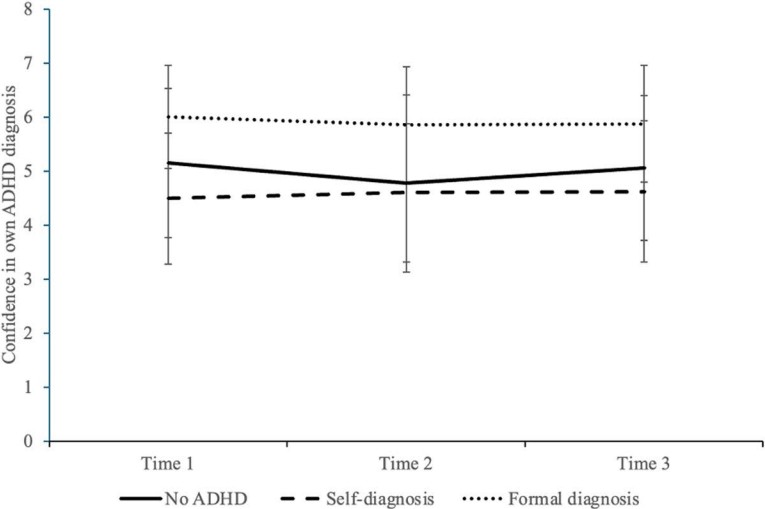

**Fig 3. Confidence in own ADHD diagnosis or lack thereof, across three time points.** The sample for this figure is participants who chose to watch the video of a psychologist describing their reasoning as to how they evaluated the TikTok videos participants watched ($N = 418$); Time 1: The beginning of the study; Time 2: After participants watched 10 TikToks about ADHD; Time 3: After participants watched a video of the psychologist rater evaluating these 10 TikToks.

ADHD and self-diagnoses groups showed variation in their confidence levels across time. At time 1 (start of study), participants in the no ADHD group ($M$ = 5.15, $SD$ = 1.38) were significantly more confident in their lack of ADHD than those with a self-diagnosis were in thinking they actually had ADHD ($M$ = 4.49, $SD$ = 1.22, $p$ < .001). However, there was no longer any difference between the two groups at time 2 (no ADHD: $M$ = 4.77, $SD$ = 1.64; self-diagnosis: $M$ = 4.60, $SD$ = 1.28; $p$ = .163). That is, those with self-diagnoses felt *more* confident that they had ADHD after watching the 10 ADHD-TikTok videos. Meanwhile, those in the no ADHD group became *less* confident they did not have ADHD after watching these same TikTok videos. As a consequence, the confidence levels of the two groups moved closer together. Finally, at time 3, participants in the no ADHD group ($M$ = 5.06, $SD$ = 1.35) were again significantly more confident than those with a self-diagnosis ($M$ = 4.62, $SD$ = 1.32; $p$ < .001). In other words, after watching the video of a psychologist explaining why the ADHD-TikTok videos were good or poor representations of the disorder, the no ADHD group became more confident that indeed, they did *not* have ADHD. However, watching the explanation video was not associated with changes in the confidence of those with an ADHD self-diagnosis that they did have ADHD.

## 6. Discussion

Young adults increasingly turn online for information and support related to mental health issues, including ADHD. Research is typically several steps behind technological progress, meaning that there are few studies about the information people receive from social media platforms and how they engage with it. Across two pre-registered studies, we aimed to assess TikTok as a psychoeducation tool for ADHD, from the perspective of both psychologists and young adults. Study 1 systematically evaluated the content, engagement, and reach of the top 100 most popular videos under the #ADHD hashtag on TikTok in January 2023. Study 2 investigated how young adults with and without self-reported ADHD appraise ADHD-related TikToks, and how their typical TikTok consumption may relate to their perceptions of ADHD.

### 6.1 Reach of #ADHD content on TikTok

ADHD-related TikTok content was extremely popular with the top 100 videos amassing almost half a billion views in total, and each video being shared and saved widely. In our study, participants with a self-diagnosis or formal diagnosis of ADHD reported watching ADHD-related TikTok content more frequently than did those without ADHD. Initial interest in such content, perhaps because users find it relevant to their lives, may be reinforced by the TikTok algorithm that learns from past user behavior to offer more videos tailored to user preferences.

The creators of the top 100 #ADHD TikTok videos regularly contributed such content; almost 80% had posted multiple videos discussing ADHD. This could be shaped by a process whereby frequent posting leads to algorithm promotion, driving more user engagement, which, in turn, further encourages frequent posting. The psychologist raters in our study found that many claims that were characterized as ADHD symptoms were transdiagnostic and that even better reflected normal human experience. Thus, some creators may focus on the breadth over the specificity of their content to attract a wider audience.

High follow counts, likes, and comments may be powerful motivators to post similar content [45]. Making videos with broad reach may make creators feel less alone, and more connected to a community of others with shared lived experiences [8,46]. Additionally, user engagement has financial benefits as TikTok pays creators based on the popularity of their content. Building an online presence also provides other financial opportunities such

as selling merch, soliciting donations, or garnering sponsors. In fact, half of the assessed video content creators stood to gain financially by selling products or asking for donations from their viewers. Crucially, we only counted explicit selling of products or solicitations for money, and did not take into account subtle advertisements. Thus, the percentage of creators with financial incentives was likely higher than 50%.

## 6.2 Evaluation of #ADHD content by psychologists and young adults

Of the content creators who listed their credentials or provided a source for their claims in the videos, most referenced their lived experience. By contrast, our study team of clinical psychologists (licensed, and in training) perceived the accuracy of the information in the TikTok videos to be low. Our finding is consistent with previous work which reported that the majority of ADHD-related content on TikTok judged by professionals to be "misleading" was posted by non-professionals [19]. It may be that some creators, drawing on their life experience, *believe* certain behaviors, thoughts, or feelings are attributable to their ADHD and depict them as such; mental health professionals, in contrast, might interpret these things as related to other aspects of creators' personalities.

Understandably, creators may be oriented to TikTok users' opinions about their content and the popularity of their content with users, as opposed to the perspectives of mental health professionals. Cynically, for some creators this could be encouraged, in part, by financial incentives. This could lead some creators to post increasingly extreme content to keep their audience's attention, similar to the "what bleeds leads" idea that guides news programming. Nonetheless, it is crucial to acknowledge that social media helps people to tell their own stories and through that, find a supportive community [7]. The users who may benefit most from seeing #ADHD TikTok videos are possibly those living with ADHD who are socially isolated, experience internalized or societal stigma about their condition, have been historically excluded from ADHD research, and lack access to treatment. Content creators may be prioritizing or catering to this group of users. The value that some users gain from watching TikTok videos, and why they seek them out, may be entirely unrelated to their scientific accuracy as judged by professionals. Thus, benefits to those users in terms of well-being—via destigmatization, feeling "seen" or understood— could be genuine. Empirically documenting such benefits is an important avenue for future research.

Compared to the psychologist raters, young adults in our sample had a more favorable view of the bottom 5 (large effect size) and a less favorable view of the top 5 psychologist-rated ADHD-TikTok videos (small effect size). At the same time, young adults' ratings for the top 5 videos were higher than those for the bottom 5 videos. Taken together, this suggests that young adults *do* critically evaluate #ADHD TikTok videos, albeit not always in a pattern that converges with psychologist judgments. Interestingly, participants with a self-diagnosis of ADHD perceived the top- and bottom-rated psychologist videos, as well as the typical TikToks they watch in their day-to-day life, more favorably than did those without ADHD; those with a self-diagnosis of ADHD also perceived the bottom-rated videos more favorably than did those with a formal ADHD diagnosis. Young adults may value the relatability, genuineness, and vulnerability of discussing one's lived experiences more than the academic background of a content creator [47] or more so than psychologists value hearing about these lived experiences. Approachable TikTok content may contrast with "colder" and harder-to-access information from empirical journal articles and clinicians [47]. In particular, people with a self-diagnosis of ADHD may have been overlooked or had their struggles minimized by psychologists, leading to healthy and understandable skepticism about mental health professionals and the formal diagnostic process.

### 6.3 Typical frequency of #ADHD TikTok consumption and perceptions of ADHD

A greater typical frequency of watching ADHD-related TikTok content was linked with more favorable evaluations of the TikToks participants see in their day-to-day life, as well as of both the top 5 and bottom 5 psychologist-rated videos. It was also associated with an increased likelihood of recommending TikTok to others as psychoeducation about ADHD. Crucially, we found these results after controlling for participant demographics and ADHD diagnostic status. These findings could suggest that, not surprisingly, people watch a greater quantity of TikToks when they perceive them to be more useful and accurate. However, another possibility is that frequent exposure to #ADHD TikTok content can normalize it so that users believe that what they see on TikTok is a typical and accurate representation of ADHD. This might also explain our finding that a greater typical frequency of watching ADHD-related TikToks was associated with estimating a higher incidence of ADHD of the general population, as well as expecting both the average person with and without ADHD to experience more ADHD symptoms in their everyday life.

The TikTok algorithm, which curates personalized recommendations based on what a user has viewed or liked before, may help users access content about ADHD. Some users who watch such content and want to understand their own experiences may conclude that they have ADHD. It is possible, given our finding that ADHD-TikTok content does not align well with the views of mental health professionals, that some people who self-diagnose based on information from TikTok would not meet the DSM-5 ADHD diagnostic criteria. At the same time, we must recognize that TikTok provides an important service to users by sharing information about ADHD, especially to those who lack financial resources or face other obstacles to seeing mental health professionals. Barriers to diagnosis and treatment disproportionally affect people with historically, persistently, or systemically marginalized identities [48]. Nonetheless, the same TikTok algorithm may allow misconceptions about ADHD to proliferate, strengthening users' beliefs in the accuracy of the information they are seeing. Prior research shows that the familiarity of repeated claims may make them seem more credible [49,50], and this may particularly happen when claims are not tempered by possible limitations. In that light, it is notable that upwards of 95% of the videos we reviewed did not acknowledge that their claims do not apply to all people with ADHD or that they may also apply to people without ADHD. Some content creators may be simply aiming to share their personal, idiosyncratic perspectives and lived experiences that they view as related to ADHD. However, a clear potential concern is that, overwhelmingly, these videos lack nuance and that alone could promote misconceptions about the condition.

### 6.4 Bridging the gap between mental health professionals and TikTok users

Half of the participants elected to watch a short video of a clinical psychologist with expertise in ADHD evaluating the ADHD-TikToks they had just seen. The decision to watch the video was framed as truly optional. Participants with ADHD (self or formally diagnosed) were more likely to watch the video compared to those without ADHD, perhaps because the topic is relevant to them. The frequency of ADHD-related TikTok consumption was not linked to their decision after controlling for diagnostic status. Overall, these results suggest that a reasonable number of young adults with ADHD are interested in hearing from mental health professionals and that consuming more TikTok content may not bias them against this information source.

We also investigated how confidence in one's ADHD diagnosis, or lack thereof, could relate to watching #ADHD TikTok videos and hearing the psychologist's views. Our results, though based on a self-selecting group interested in watching the psychologist video, revealed a general trend. First, the confidence of participants with a formal ADHD diagnosis remained unchanged after both the ADHD-related TikTok videos, and after the psychologist's evaluation of them. Conversely, watching ADHD-related TikTok videos may have been associated with some self-diagnosed individuals and some people *without* ADHD becoming more likely to think they do have ADHD. In turn, watching the psychologist's explanation video may have reassured those without ADHD that they do not have it, but was not associated with changes in the confidence of the self-diagnosed group. Possibly, while the impact on those with self-diagnoses or formal diagnoses appeared limited, a psychologist's tempering remarks seemed to reduce concerns raised by watching #ADHD TikTok videos in those with no ADHD history, suggesting that more online ADHD content from licensed mental health professionals may be warranted.

## 6.5  Limitations and future directions

In Study 1, our sample consisted of only the 100 most popular TikTok videos, two of which were excluded from the analysis. In addition, our database was built based on TikTok's search engine which excludes videos that contain ads, but some such videos are widely popular. The platform also does not allow for the search of deleted videos, regardless of their popularity at their peak. Since we only looked at the most popular videos that our results may not generalize to less popular ADHD-related content. Furthermore, often users engage with and trust content from creators they follow more than they do the "viral" video of the week. Future research should examine the accuracy and quality of the videos made by identified, popular ADHD content creators. Finally, although we captured the engagement of users based on their comments, we did not assess if the comments were mostly positive, neutral, or negative. "Rage-bait" content is designed to elicit (mostly negative) responses from the viewers who are compelled to engage with it to express their disapproval. Although, at face value, no such content was detected here, it is possible that for some of the videos the engagement captured was not necessarily all positive and at least some of the comments were critical.

In Study 2, participants were university students enrolled in an introductory psychology course. Therefore, they may have a different view of ADHD and mental health than non-students, or the general population. In addition, our sample predominantly consisted of women, which is the case for most studies that recruit undergraduates from a psychology human subjects pool. ADHD is estimated to affect a near equal ratio of men to women in emerging adulthood. In addition, women with ADHD are more likely than men to be misdiagnosed (or not seen) by professionals (Faheem et al., 2022). Our study may be highly pertinent to women, as this group may turn to TikTok for ADHD information and diagnose themselves precisely because they are not recognized by professionals. Nonetheless, we do not know the extent to which the gender imbalance in our study restricts the generalizability of our results to the population of emerging adults who are formally diagnosed or self-diagnosed with ADHD.

All study measures were based on participant self-report, which could be influenced by social desirability bias or memory recall errors. Notably, we did not assess or confirm participants' ADHD diagnoses. Third, the time gap between viewing ADHD-TikTok videos/ the psychologist explanation video and subsequent questions about their confidence in their own ADHD may have been insufficient for detecting any lasting associations. Moreover, we captured the videos for Study 1 in January of 2023 but participants in Study 2 watched them in fall 2023 or winter 2024. This means that some participants may have already seen those videos, given their popularity. Alternatively, some of the videos may have referenced outdated

trends. Finally, future research should explore the aspects of TikTok psychoeducation content that participants find helpful to capture a more well-rounded view of the platform.

Prior research shows that people are more likely to engage with and trust news sources they use frequently [51]. Similarly, it is possible that users engage with and trust content from creators they recognize, watch regularly, and follow, more so than users do with the "viral" video of the week. Future research should examine whether young adults trust mental health content on social media more when it comes from creators they follow, compared to "viral" videos from influencers they do not follow

## 6.6 Implications and conclusion

This work provides a starting point for understanding depictions of ADHD on TikTok.

Generally, our results highlight the power of social media to shape public understanding of mental health concerns like ADHD. On a positive note, this underscores the importance of TikTok for democratizing mental health information, and for promoting understanding and destigmatization of the challenges faced by those with ADHD. At the same time, TikTok's anecdotal content could lead some viewers to misattribute normal behaviors or those better explained by other conditions to be signs of ADHD, complicating an already challenging differential diagnosis and treatment process.

Mental health professionals may experience patients seeking an ADHD diagnosis or who are self-diagnosed based on information from TikTok. It may be important for professionals to listen to patients' experiences with ADHD information on social media and to hear about what patients have found valuable in this information. At the same time, mental health professionals can attempt to conduct a thorough assessment that includes history and multiple informants, that balances anecdotal experience whenever possible. They can take the time to explain the diagnostic criteria for ADHD, to demystify the diagnostic process and ensure it is collaborative. Mental health professionals also need to be aware that in the absence of easily accessible or relatable information people may acquire their ADHD knowledge from others online, likely based mainly on anecdotal or personal experiences. Therefore, it may behoove professionals to advocate for more equitable access to ADHD diagnosis and treatment, and ADHD stigma reduction efforts. It may be that such efforts can help to address the discrepancy between mental health professionals' and TikTok users' perceptions of ADHD, which ultimately will help better serve those with related needs in the broad population.

## Supporting information

**S1 Appendix. Descriptive statistics of study variables for non-binary, two-spirit, agender, questioning, and participants who chose not to disclose their gender identity (N = 49).** (DOCX)

## Author contributions

**Conceptualization:** Vasileia Karasavva, Caroline Miller, Nicole Groves, Will Canu, Amori Mikami.

**Formal analysis:** Vasileia Karasavva.

**Methodology:** Vasileia Karasavva, Caroline Miller.

**Project administration:** Vasileia Karasavva.

**Supervision:** Will Canu, Amori Mikami.

**Visualization:** Vasileia Karasavva.

**Writing – original draft:** Vasileia Karasavva, Caroline Miller, Nicole Groves, Andrés Montiel, Amori Mikami.

**Writing – review & editing:** Vasileia Karasavva, Caroline Miller, Nicole Groves, Andrés Montiel, Will Canu, Amori Mikami.

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
