## [Decision Letter · Decision Letter 0]

1 Dec 2024

PONE-D-24-37728A double-edged hashtag: Evaluation of #ADHD-related TikTok content and its associations with perceptions of ADHDPLOS ONE

Dear Dr. Karasavva,

Thank you for submitting your manuscript to PLOS ONE. After careful consideration, we feel that it has merit but does not fully meet PLOS ONE’s publication criteria as it currently stands. Therefore, we invite you to submit a revised version of the manuscript that addresses the points raised during the review process.

In light of the reviewer’s suggestions, I invited you to make a revision. Particularly, provide additional information about the accounts used to collect TikTok videos. Whose accounts did you use? Also, please explain why the gender gap is so significant in your analysis.

We look forward to receiving your revised manuscript.

Kind regards,

Chang Sup Park, Ph.D.

Academic Editor

PLOS ONE

2. In your Methods section, please include additional information about your dataset and ensure that you have included a statement specifying whether the collection and analysis method complied with the terms and conditions for the source of the data.

3. For studies involving third-party data, we encourage authors to share any data specific to their analyses that they can legally distribute. PLOS recognizes, however, that authors may be using third-party data they do not have the rights to share. When third-party data cannot be publicly shared, authors must provide all information necessary for interested researchers to apply to gain access to the data. (https://journals.plos.org/plosone/s/data-availability#loc-acceptable-data-access-restrictions) For any third-party data that the authors cannot legally distribute, they should include the following information in their Data Availability Statement upon submission: 1) A description of the data set and the third-party source 2) If applicable, verification of permission to use the data set 3) Confirmation of whether the authors received any special privileges in accessing the data that other researchers would not have 4) All necessary contact information others would need to apply to gain access to the data

4. Please amend your authorship list in your manuscript file to include author Vasileia Karasavva, Caroline Miller, Nicole Groves, Andrés Montiel, Will Canu, Amori Mikami.

5. Please amend your list of authors on the manuscript to ensure that each author is linked to an affiliation. Authors’ affiliations should reflect the institution where the work was done (if authors moved subsequently, you can also list the new affiliation stating “current affiliation:….” as necessary).’

7. Please include a separate caption for each figure in your manuscript.

Additional Editor Comments:

In light of the reviewer’s suggestions, I invited you to make a revision. Particularly, provide additional information about the accounts used to collect TikTok videos. Whose accounts did you use? Also, please explain why the gender gap is so significant in your analysis.

Reviewers' comments:

Reviewer's Responses to Questions

**Comments to the Author**

1. Is the manuscript technically sound, and do the data support the conclusions?

Reviewer #1: Yes

2. Has the statistical analysis been performed appropriately and rigorously? 

Reviewer #1: Yes

3. Have the authors made all data underlying the findings in their manuscript fully available?

Reviewer #1: Yes

4. Is the manuscript presented in an intelligible fashion and written in standard English?

Reviewer #1: Yes

5. Review Comments to the Author

Reviewer #1: This paper offers timely and relevant insight into the relationship between ADHD content shared on TikTok and user’s perceptions of that information. While there has been a growing body of literature in recent years which theorizes about identity, algorithms, and mental health, I am unaware of a study like this one. The value this brings to the understanding of mental health TikTok is systematic, statistics-driven evidence that non-expert TikTok users view psychoeducational content on the app differently than experts. In my opinion, the most interesting findings are the differences between self-diagnosed and formally diagnosed participants in their views of the disorder and the quality of information about ADHD on TikTok.

As far as I can judge, the paper is technically sound and the conclusions seem reasonable given the evidence. It appears that careful thought has been given to both the collection and analysis of data. I have a few minor questions/comments for the authors before the publication of this article which can broadly be categorized as questions about reporting of methodology and about referencing/engaging with relevant literature in the introduction and conclusion. These can be thought of mostly as points of clarity and polish.

Methodology

Section 2.1; Page 9: I wonder if the authors could provide additional information about the account(s) used to collect TikTok videos. I have reason to believe that search results can vary based on the user (i.e., even if two users input the same hashtag into the search bar and filter by most-viewed, the algorithm will show them different videos). It’s unclear if this is merely anecdotal, but it may be worth noting whose account was used or if a more objective form of data scraping was used.

Section 3.1; Page 13: The authors write, “50% of the content creators promoted products”. Did the researchers take into account subtle advertisements or does this figure only represent explicit ads? E.g., sometimes a creator will share their “ADHD routine”, and one item is subtly listed for sale on their account. They never mention that they’re selling it, but it is still an ad of some kind.

Section 4.1, Page 15: Why is the gender gap so significant between men and women? Given that ADHD is still considered to be more prevalent in men vs. women, do the authors feel their participant pool is unrepresentative of the general population in any meaningful way?

Section 4.3.3, Page 19(6): “Always” is a curious item for viewing online content since it is unlikely that anyone always views one thing online compared to others. Did participants select this option?

Section 5.1, Page 21(1): Beyond the obvious reason that it would make the comparison more complicated, is there a reason to exclude non-binary participants from the analysis?

Referencing

The authors note a couple of times that there is very little research on TikTok, and that may be true within the clinical psychology boundary, but I’d like to recommend that the authors explore literature in media theory, information sciences, disability studies, and philosophy of psychiatry that may enrich some of the claims made in the introduction and discussion sections of this paper. Some examples:

Page 3(7-8): “Social media centres the perspectives of those with lived experience…” may be beneficial to review research on neurodiversity/neurodivergence in online spaces for citation here (e.g., Arnaud & Gangé-Julien, 2023, https://doi.org/10.1080/09515089.2023.2174425).

Page 3(23): The citation here is from 2021. Given the speed with which TikTok has dominated the landscape, I’d be interested to know if it is still true that it is the “least studied among major social media platforms”.

Page 4(5-9): Again, may be valuable to cite some scholarship about the value of digital tools in disability communities.

Page 4(13): Would recommend citing something about how the TikTok algorithm works here.

Page 6(12): “Mental health…content…could influence users’…” There have recently been some pieces in social media journals on the relationship between users’ perceptions of identity and use of social media (e.g., Leveille, 2024, https://doi.org/10.1177/20563051241269260). Also, e.g., scholarship on movements in participatory psychiatry.

Page 34: The authors make claims about what motivates creators to post the kind of content that they do, which could benefit from some engagement in literature on social media incentives etc.

Page 39(1): The claim that users trust creators that they follow more than viral videos could benefit from a citation. If there isn’t one, then this would be an interesting area of study. In the case of TikTok, I’m not sure if people do prioritize who they follow, since the FYP is a prominent source of content.

Misc.

Finally, on Page 7, the authors reference the prevalence of anti-vax content online as motivation for reviewing the factuality of ADHD videos. I’m a little concerned about the comparison this draws between anti-vax communities and mental health communities. While anti-vax creators often promote a dangerous, potentially lethal ideology that has no basis in medical science, it’s not clear that ADHD creators who similarly share misinformation are: 1. Capable of causing that kind of harm, or 2. Wrong to share content that fails to align with DSM criteria. I.e., there are valid criticisms about the DSM and mental health communities could be spaces for positive change. I’m not sure the same can be said for Anti-vaxxers.

This is sort of what distinguishes misinformation (which could be harmless or accidental) and disinformation (which is harmful and concerted). The kinds of epistemic harm found online are diverse, and I think it's important to note that nuance.

6. PLOS authors have the option to publish the peer review history of their article (what does this mean? ). If published, this will include your full peer review and any attached files.

**Do you want your identity to be public for this peer review?** For information about this choice, including consent withdrawal, please see our Privacy Policy .

Reviewer #1: No

---

## [Author Response · Author response to Decision Letter 0]

24 Dec 2024

Comments to the Author

1. Is the manuscript technically sound, and do the data support the conclusions?

Reviewer #1: Yes

2. Has the statistical analysis been performed appropriately and rigorously?

Reviewer #1: Yes

3. Have the authors made all data underlying the findings in their manuscript fully available?

Reviewer #1: Yes

4. Is the manuscript presented in an intelligible fashion and written in standard English?

Reviewer #1: Yes

5. Review Comments to the Author

Reviewer #1: This paper offers timely and relevant insight into the relationship between ADHD content shared on TikTok and user’s perceptions of that information. While there has been a growing body of literature in recent years which theorizes about identity, algorithms, and mental health, I am unaware of a study like this one. The value this brings to the understanding of mental health TikTok is systematic, statistics-driven evidence that non-expert TikTok users view psychoeducational content on the app differently than experts. In my opinion, the most interesting findings are the differences between self-diagnosed and formally diagnosed participants in their views of the disorder and the quality of information about ADHD on TikTok.

As far as I can judge, the paper is technically sound and the conclusions seem reasonable given the evidence. It appears that careful thought has been given to both the collection and analysis of data. I have a few minor questions/comments for the authors before the publication of this article which can broadly be categorized as questions about reporting of methodology and about referencing/engaging with relevant literature in the introduction and conclusion. These can be thought of mostly as points of clarity and polish.

Response. We appreciate your praise of this paper, as well as your helpful suggestions for improvement. You have raised excellent points and we respond to each one, below.

Methodology

R1.1. Section 2.1; Page 9: I wonder if the authors could provide additional information about the account(s) used to collect TikTok videos. I have reason to believe that search results can vary based on the user (i.e., even if two users input the same hashtag into the search bar and filter by most-viewed, the algorithm will show them different videos). It’s unclear if this is merely anecdotal, but it may be worth noting whose account was used or if a more objective form of data scraping was used.

Response. Thank you for bringing this up. You are correct that a user’s engagement history affects what content they are served on TikTok. In order to try to collect videos the most objectively, we created a brand new TikTok account for this study. We then used the search bar to find videos under the hashtag #ADHD, which we sorted by view count. We recognize the possibility that the results we were served might not be exactly the same as what other users would see if they conducted the same search. We have edited our Method to include this information.

In section 2.1. Video search:

“We created a new TikTok account for the study and queried the hashtag “#ADHD” using the search bar. We sorted the search results by view count and screen recorded the 100 most viewed videos on a single day: January 10, 2023. Notably, the TikTok algorithm offers users videos based on their past engagement on the app, so searching the #ADHD hashtag might show different results to different users. However, because we focused on the most popular videos, our sample of 100 videos likely represents the typical content users see.”

R1.2. Section 3.1; Page 13: The authors write, “50% of the content creators promoted products”. Did the researchers take into account subtle advertisements or does this figure only represent explicit ads? E.g., sometimes a creator will share their “ADHD routine”, and one item is subtly listed for sale on their account. They never mention that they’re selling it, but it is still an ad of some kind.

Response. This is a very good point. We only counted explicit promotions or sales of products, or explicit solicitations for money. However, it is likely that there were other more subtle advertisements that we missed. We now describe how we coded videos for financial incentives in more detail. In addition, we mention the possibility that we missed more subtle forms of advertising as a limitation in our Discussion.

In section 2.2. Video metrics:

“[…] and any apparent potential for financial gain (e.g., selling products related to the diagnosis, treatment, or management of ADHD, links to Venmo or Cashapp accounts). This was assessed by noting explicit selling of products related to the diagnosis, treatment, or management of ADHD, or explicit solicitations for money, in the included videos or through links to Venmo or Cashapp accounts on the creators’ profiles.”

In section 6.1. Reach of #ADHD Content on TikTok

“Crucially, we only counted explicit selling of products or solicitations for money, and did not take into account subtle advertisements. Thus, the percentage of creators with financial incentives was likely higher than 50%.”

R1.3. Section 4.1, Page 15: Why is the gender gap so significant between men and women? Given that ADHD is still considered to be more prevalent in men vs. women, do the authors feel their participant pool is unrepresentative of the general population in any meaningful way?

Response. Our sample consisted of undergraduate students at a university in Canada who were enrolled in a psychology course and completed our study for partial course credit. Samples like these tend to overwhelmingly consist of women. We understand why you raised a concern about the generalizability of the sample, especially because this is a study about ADHD. Interestingly, while ADHD is estimated to affect 2-3 times as many boys than girls in childhood, the gender ratio in emerging adulthood is near equal across men and women. In addition, professionals are more likely to misdiagnose (or fail to diagnose) ADHD in women compared to in men, meaning that reports of ADHD prevalence rates likely underestimate women with the condition. It also means that our study, which intentionally oversampled people with self-diagnoses of ADHD, may be highly pertinent to women – as this group may turn to TikTok for ADHD information precisely because their diagnosis is not recognized by professionals. Adult women with ADHD are at least as impaired as (or possibly more impaired than) men with ADHD - suggesting that women’s underdiagnosis by professionals is concerning and that women do need ADHD resources and treatment. (For reviews of gender differences in adults with ADHD, see Faheem et al. [2022], Williamson & Johnston [2016]). We appreciate the opportunity to elaborate on the gender differences in ADHD prevalence rates and to speculate about the representativeness of our sample. We have included some of these thoughts in the Limitations section of the manuscript.

“In Study 2, participants were university students enrolled in an introductory psychology course. Therefore, they may have a different view of ADHD and mental health than non-students, or the general population. In addition, our sample predominantly consisted of women, which is the case for most studies that recruit undergraduates from a psychology human subjects pool. ADHD is estimated to affect a near equal ratio of men to women in emerging adulthood. In addition, women with ADHD are more likely than men to be misdiagnosed (or not seen) by professionals (Faheem et al., 2022). Our study may be highly pertinent to women, as this group may turn to TikTok for ADHD information and diagnose themselves precisely because they are not recognized by professionals. Nonetheless, we do not know the extent to which the gender imbalance in our study restricts the generalizability of our results to the population of emerging adults who are formally diagnosed or self-diagnosed with ADHD.”

R1.4. Section 4.3.3, Page 19(6): “Always” is a curious item for viewing online content since it is unlikely that anyone always views one thing online compared to others. Did participants select this option?

Response. We see your point. We were thinking about this more in the context of a typical TikTok scrolling session. In other words, if each time you open the app you are presented at some point with ADHD-related material, then “always” would be appropriate. Participants did select that option – although it was the least popular choice by far. For example, 4% (N = 54) selected “always” for the item “How often does material about ADHD come up on your ForYou or Following page without you searching for it”. Thank you for pointing this out. Likely, “all the time” would have been a better option instead of “always”. We will make this change when using this scale in the future.

R1.5 Section 5.1, Page 21(1): Beyond the obvious reason that it would make the comparison more complicated, is there a reason to exclude non-binary participants from the analysis?

Response. To clarify, non-binary and gender diverse participants were only excluded from the analyses that examined gender, but were included in all other analyses. We did give a lot of thought to how to handle these participants, and we consulted with a colleague whose main research questions involve sex and gender. Based on our colleague’s advice, we excluded these participants because we did not recruit enough non-binary, two-spirit, agender, agender, and questioning people to have enough statistical power to make any meaningful comparisons. Instead, as recommended by our colleague, we included the descriptive statistics for all study variables of these participants in Table A1 in the Appendix. This is so that other researchers might be able to use this data if, for instance, they are pooling multiple samples to study non-binary and gender diverse people.

Section 5.1.

“Non-binary, two-spirit, and agender people, as well as those questioning their gender identity and those who chose not to disclose their gender (n = 49), were not included in the gender analyses. However, these participants were included in all other analyses. A table with their descriptive statistics on all study variables is in Appendix A.”

R1.6. The authors note a couple of times that there is very little research on TikTok, and that may be true within the clinical psychology boundary, but I’d like to recommend that the authors explore literature in media theory, information sciences, disability studies, and philosophy of psychiatry that may enrich some of the claims made in the introduction and discussion sections of this paper. Some examples:

Page 3(7-8): “Social media centres the perspectives of those with lived experience…” may be beneficial to review research on neurodiversity/neurodivergence in online spaces for citation here (e.g., Arnaud & Gangé-Julien, 2023, https://doi.org/10.1080/09515089.2023.2174425).

Response. Thank you for the suggestion. We have added the recommended work to our paper. We have also added several other papers based on your other suggestions (See R1.8, R1.9, R1.10, R1.11, R1.12).

R1.7. Page 3(23): The citation here is from 2021. Given the speed with which TikTok has dominated the landscape, I’d be interested to know if it is still true that it is the “least studied among major social media platforms”.

Response. We believe so. Even though it has been 3 years since the referenced paper came out, research moves slow and social media platforms like Facebook and Instagram have been studied for many years already and continue to be. We tried to get a sense of the current status of the literature by searching the names of major social media platforms on Google Scholar. We chose Google Scholar because it casts a slightly wider net than other tools like Web of Science or PsycInfo and includes more works that are outside of peer-reviewed, published journal articles. As can be seen in the table below, the research on TikTok still appears to be behind that of other platforms at the moment. It is possible that if TikTok continues to rise in popularity (especially among younger people), this will change in a few years!

Platform Approximate number of articles (all time) Approximate number of articles (since 2020)

TikTok 2.3 Million 206,000

Facebook 7.8 Million 2.1 Million

Instagram 4.9 Million 663,000

Twitter 8.2 Million 2.1 Million

R1.8. Page 4(5-9): Again, may be valuable to cite some scholarship about the value of digital tools in disability communities.

Response. This is a very helpful recommendation. We have incorporated the following literature:

Sweet, K. S., LeBlanc, J. K., Stough, L. M., & Sweany, N. W. (2020). Community building and knowledge sharing by individuals with disabilities using social media. Journal of Computer Assisted Learning, 36(1), 1-11.

Danker, J., Strnadová, I., Tso, M., Loblinzk, J., Cumming, T. M., & Martin, A. J. (2023). ‘It will open your world up’: The role of mobile technology in promoting social inclusion among adults with intellectual disabilities. British Journal of Learning Disabilities, 51(2), 135-147.

R1.9. Page 4(13): Would recommend citing something about how the TikTok algorithm works here.

Response. We have added the following two citations:

Kang, H., & Lou, C. (2022). AI agency vs. human agency: understanding human–AI interactions on TikTok and their implications for user engagement. Journal of Computer-Mediated Communication, 27(5), zmac014.

Siles, I., Valerio-Alfaro, L., & Meléndez-Moran, A. (2024). Learning to like TikTok... and not: Algorithm awareness as process. New Media & Society, 26(10), 5702-5718.

R1.10. Page 6(12): “Mental health…content…could influence users’…” There have recently been some pieces in social media journals on the relationship between users’ perceptions of identity and use of social media (e.g., Leveille, 2024,

https://doi.org/10.1177/20563051241269260). Also, e.g., scholarship on movements in participatory psychiatry.

Response. Thank you for your suggestions. We have added this citation and a couple more that were published very recently, after we submitted the previous version of our manuscript:

Leveille, A. D. (2024). “Tell me you have ADHD without telling me you have ADHD”: Neurodivergent identity performance on TikTok. Social Media+ Society, 10(3), 20563051241269260.

Foster, A., & Ellis, N. (2024). TikTok-inspired self-diagnosis and its implications for educational psychology practice. Educational Psychology in Practice, 40(4), 1-18.

Lovelace, S. (2024). The relationship between TikTok use and self-diagnosis of ADHD: Exploring the influence of mental health videos on diagnosis-seeking behavior (Master's thesis, Harvard University).

R1.11. Page 34: The authors ma

---

## [Decision Letter · Decision Letter 1]

31 Jan 2025

A double-edged hashtag: Evaluation of #ADHD-related TikTok content and its associations with perceptions of ADHD

PONE-D-24-37728R1

Dear Dr. Karasavva,

We’re pleased to inform you that your manuscript has been judged scientifically suitable for publication and will be formally accepted for publication once it meets all outstanding technical requirements.

Kind regards,

Chang Sup Park, Ph.D.

Academic Editor

PLOS ONE

Additional Editor Comments (optional):

In light of the reviewers' positive comments, we are glad to accept your paper.

Reviewers' comments:

Reviewer's Responses to Questions

**Comments to the Author**

1. If the authors have adequately addressed your comments raised in a previous round of review and you feel that this manuscript is now acceptable for publication, you may indicate that here to bypass the “Comments to the Author” section, enter your conflict of interest statement in the “Confidential to Editor” section, and submit your "Accept" recommendation.

Reviewer #1: All comments have been addressed

2. Is the manuscript technically sound, and do the data support the conclusions?

Reviewer #1: Yes

3. Has the statistical analysis been performed appropriately and rigorously? 

Reviewer #1: Yes

4. Have the authors made all data underlying the findings in their manuscript fully available?

Reviewer #1: Yes

5. Is the manuscript presented in an intelligible fashion and written in standard English?

Reviewer #1: Yes

6. Review Comments to the Author

Reviewer #1: (No Response)

7. PLOS authors have the option to publish the peer review history of their article (what does this mean? ). If published, this will include your full peer review and any attached files.

**Do you want your identity to be public for this peer review?** For information about this choice, including consent withdrawal, please see our Privacy Policy .

Reviewer #1: No

---

## [Editor Report · Acceptance letter]

PONE-D-24-37728R1

PLOS ONE

Dear Dr. Karasavva,

I'm pleased to inform you that your manuscript has been deemed suitable for publication in PLOS ONE. Congratulations! Your manuscript is now being handed over to our production team.

Kind regards,

on behalf of

Dr. Chang Sup Park

Academic Editor

PLOS ONE